# Genetic Diversity Analysis and Core Germplasm Collection Construction of *Camellia oleifera* Based on Fruit Phenotype and SSR Data

**DOI:** 10.3390/genes13122351

**Published:** 2022-12-13

**Authors:** Yunzheng Zhu, Deyang Liang, Zejun Song, Yi Tan, Xiaolan Guo, Delu Wang

**Affiliations:** 1College of Forestry, Guizhou University, Guiyang 550025, China; 2College of Life Sciences, Huizhou University, Huizhou 516007, China

**Keywords:** *Camellia oleifera*, phenotype, molecular marker, core germplasms collection, PowerCore

## Abstract

Many *Camellia oleifera* germplasm resources were collected from Guizhou Province, but the fruit morphological variation and genetic diversity of *C. oleifera* germplasm resources remain unclear. The genetic diversity of *C. oleifera* germplasms resources in Guizhou was studied based on fruit traits and simple sequence repeat (SSR) molecular markers to build a core collection. This paper aims to provide a scientific basis for the collection, management, development, and utilization of *C. oleifera* resources in Guizhou province. The variation coefficients among and within varieties of seven fruit phenotypic traits of *C. oleifera* ranged from 11.79% to 61.76% and from 8.15% to 42.31%, respectively, showing rich phenotypic variation. Furthermore, 12 SSR markers were used to analyze the genetic diversity. These primers generated 214 polymorphic bands, and the average number was 17.833. The average number of effective alleles (Ne), Shannon’s information index (I), observed heterozygosity (Ho), expected heterozygosity (He), polymorphic information content (PIC), and major allele frequency (MAF) were 8.999, 2.301, 0.965, 0.50, 0.836, and 0.238, respectively. The results showed that 12 SSR markers had high polymorphism, and the genetic diversity of 167 *C. oleifera* germplasm resources was high. Based on SSR molecular marker information and fruit traits clustering, 167 *C. oleifera* germplasm resources were divided into three groups. When constructing core collections based on fruit traits and molecular marker information, the PowerCore-25 of core collections greatly preserves fruit traits and improves genetic diversity. This paper can provide a reference for the genetic diversity and fruit traits variation of *C. camellia* germplasm resources in Guizhou Province. It is significant for establishing a core collection, thus promoting germplasm innovation and the development of the oil tea industry in Guizhou.

## 1. Introduction

*C. oleifera*, belongs to the genus *Camellia* of the Teaceae family [1]. *C. camellia*, olive (*Olea europaea*), oil palm (*Elaeis guineensis* Jacq.), and coconut (*Cocos nucifera* L.) are four important woody oil plants around the world [2]. *C. oleifera* has high genetic diversity and phenotypic variation, and the study of genetic diversity is the basis of breeding better varieties. Therefore, understanding the genetic diversity of *C. oleifolia* and constructing a core germplasm resources bank of *C. oleifolia* is an important link in the development of local *C. oleifolia*.

Genetic diversity is the basis of biological diversity and is the driving force for the stability and continuous evolution of species. The methods for studying genetic diversity include morphology, cytology, and molecular marker technology. Morphology, as the most simple and convenient labeling method, can reveal the degree of genetic variation to a certain extent and explore potential target traits [3]. With the development of science and technology, molecular marker technology has become one of the main ways to study genetic diversity, population structure, and kinship. SSR molecular markers, due to their high polymorphism and good stability, are currently considered one of the most important molecular markers. This method is widely used to study the genera *Pyrus* [4], *Juglans mandshurica Maxim* [5], Tunisian melon (*Cucumis melo* L.) [6], and pumpkins (*Cucurbita* spp.) [7]. It is also used to study camellia oil. He et al. [8] studied the germplasm resources of 150 oil teas; the Dice genetic similarity coefficients of 150 germplasm materials ranged from 0.05 to 0.91, which demonstrates rich genetic diversity. Both morphological and molecular markers can be used to construct a core collection. The construction of core collections based on morphological, physiological, and biochemical traits has been reported for pomegranate (*Punica granatum* L.) [9], Huangxinzimu (*Catalpa fargesii f.duclouxii*) [10], and *Ziziphus mauritiana* Lam. [11]. Based on molecular markers, core germplasm collection groups of *Phaseolus lunatus* [12], Myrtaceae (*Eucalyptus cloeziana* F. Muell.) [13], hazelnut (*Corylus avellana* L.) [14], *Akebia trifoliata* (*Thunb.*) Koidz [15], and *Prunus sibirica* [16] were constructed. In *Pinus yunnanensis* Franch. [17], *Perilla* [18], and *Cymbopogon winterianus* Jowitt [19], the diversity of phenotypes, agronomic traits, and molecular marker information of core collections has been verified. Furthermore, the degree of variation and genetic diversity of the core collection should be increased as far as possible.

This paper aims to establish the variation in fruit traits of *C. oleifolia* and evaluate the genetic diversity and relationships of *C. oleifera* germplasm resources in Guizhou Province. Constructing core collections of fruit traits and molecular marker information lays the foundation for the breeding of *C. oleifolia* and the protection of germplasm resources.

## 2. Materials and Methods

### 2.1. Plant Material

The experimental samples were collected from key oil tea distribution areas in Guizhou Province from 2008 to 2011. Information about the collection area is shown in Table 1. All of the samples were excellent families, clones, and superior varieties approved (recognized) by Guizhou Province and were over 10 years old (detailed information can be found in Attached Table 1). Leaves were quickly stored in dry ice, and promptly brought back to the laboratory in a −80 °C ultra-low-temperature refrigerator, as DNA extraction materials.

### 2.2. Fruit Phenotype Determination

The index of phenotypic and economic traits was determined using the method described by Yang et al. [20]. In the ripening period, three plants of each variety were randomly selected to collect a sufficient amount of fruit in a mesh bag and marked. The fruits were spread on a shelf and measured immediately. A total of 20 fruits were randomly selected from each plant for determination, and vernier calipers were used to measure the fruit vertical diameter (FVD), fruit horizontal diameter (FHD), and peel thickness (PT). Single fresh fruit weight (FFW) and fresh seed weight (FSW) were determined using an electronic balance. Then, measured peel thickness (PT) was measured and the number of seeds (SGN) recorded. After the phenotypic data had been collected, the fresh weight of the seeds of 20 fruits from the same tree was measured and the seeds were then dried in an oven at 80 °C. The dry seed weight and dry kernel weight were determined using an electronic balance. The oil content of the dry kernels (DKOC) was determined by means of the cable extraction method, with petroleum ether (60–90 °C) as the solvent, and extracted for 6 h in the fat detector (temperature set at 80 °C). Before pumping, the kernel powder mass (m) was weighed. Kernel powder was put into the filter paper package to weigh the first mass (m_1_). After extraction, it was baked in an oven at 105 °C for 1 h to weigh the second mass (m_2_).
Fresh seed rate (FSR, %) = fresh seed weight/ fresh fruit weight × 100%.(1)
Dry seed rate (DSR, %) = dry seed weight/fresh fruit weight × 100%.(2)
Dry kernel yield (DKR, %) = dry kernel weight/dry seed weight × 100%.(3)
DKOC (%) = (m_1_ − m_2_)/m(4)

### 2.3. DNA Extraction and PCR Analysis

The SSR molecular marker test includes DNA extraction, primer screening, fluorescence quantitative PCR, and electrophoresis tube detection. DNA was extracted using a genomic kit from Beijing Botanical Biofet Plant. This DNA extraction kit was used to select primers as the 36 main reference pairs [21,22,23], and 12 pairs of primers with high and stable polymorphism were screened (Table 2) for fluorescence quantitative PCR and electrophoresis tube detection.

The fluorescence quantitative PCR consists of a 20 µL system, including 17 µL of gold mix (green), 1 µL of front primers, 1 µL of back primers, and 1 µL of DNA template. The amplification procedures are listed as follows. First, pre-denaturation was performed at 98 °C for 2 min. The second stage was the cycle stage. Samples were denatured at 98 °C for 10 s, annealing at 55–62 °C for 10 s, extended at 72 °C for 10 s, and cycled 35 times. The third stage was the extension stage. Samples were extended at 72 °C for 5 min. For the capillary test, the mixing plate was heated at 95 °C for 5 min with a metal bath heater and immediately put into an ice box at −20 °C. The mixing plate was removed after cooling, centrifuged at 4000 rpm, thawed, and mixed well. Then, it was placed in an ABI 3730xl sequencer for capillary electrophoresis.

### 2.4. Construction of Core Collection

The core collection was constructed with different sampling ratios (10%, 15%, 20%, 25%, and 30%). QGA software adopts Euclidean distance, multiple clustering priority sampling methods, and the shortest clustering method. PowerCore is an M strategy with a heuristic search for establishing core sets [26]. The core collections were evaluated using genetic diversity parameters and effective value parameters of fruit traits, and the difference significance was detected by *t*-test.

### 2.5. SSR Analysis and Statistical Analysis

The exact loci and statistic genotyping data were analyzed with Gene Mapper 4.1. The genetic diversity indexes of SSR loci, including the number of different alleles (Na), number of effective alleles (Ne), Shannon’s information index (I), observed heterozygosity (Ho), expected heterozygosity (He), polymorphism information content (PIC), and major allele frequency (MAF) were calculated using GenAlEx 6.51 and PowerMarker software. Genetic structure and phylogenetic trees were constructed by means of Structure 2.3.4 and MEGA v.6. The quantitative traits were divided into 10 levels according to mean value (X) and standard deviation (σ), and Xi < X − 2σ and Xi > X + 2σ increased from 1 to 10 [10]. The phenotypic differentiation coefficient was calculated as VST = δ2t/s/(δ2t/s + δ2s) × 100%, where δ2t/s and δ2s were variances between and within populations, respectively [27]. Other data analysis was completed using SPSS 26, and variance square calculation was completed using Minitab 19.

## 3. Results

### 3.1. Analysis of Fruit Phenotypic Variation

Variance analysis was conducted on seven phenotypic traits of 167 *C. oleifera* germplasm resources in Guizhou province. The F-value test showed that the fruit phenotypic traits exhibited significant differences within and among varieties (Table 3). These results indicate the rich diversity in the phenotypes of *C. oleifera*.

The phenotypic traits of *C. oleifera* differed greatly between and within varieties, and the variation between varieties was greater than that within varieties. The variation coefficients among different varieties and within the varieties were in the range of 11.79–61.76% and 8.15–42.31%, respectively. The highest variation among varieties was in FFW, and the lowest was the fruit shape index (FSI). The highest variation within varieties was SGN, and the lowest was in FSI. The variation ranges of variance components among different varieties and within the varieties were 0.006–15.108 and 0–2.294, respectively. The trait with the largest variance component among different varieties was PT (67.35%), and the one with the smallest was SGN (43.68%). The phenotypic differentiation coefficients ranged from 80.15% to 95.56%, and the phenotypic differentiation coefficients of fruit traits were greater than 70%. The results indicate that the fruit traits mainly vary among different varieties, which is consistent with the results of the coefficient of variation (Table 4).

### 3.2. Correlation and Classification of Fruit Phenotype and Economic Characters

The correlation of 12 fruit traits of 167 *C. oleifera* germplasm resources was highly significant. Most of the phenotypic traits were positively correlated, and the economic traits were positively correlated with each other. However, most phenotypic traits were negatively correlated with economic traits (Figure 1).

In the cluster heat map, the traits are divided into two groups. The first group consists of economic traits, including FSR, DSR, DKR, and DKOC. The second group comprises fruit phenotypic traits and HGW. In the second group, FVD and PT are divided into one class, FSW and HGW are divided into one class, and the rest are divided into one class (Figure 2).

The 167 germplasm resources were divided into three groups according to 12 traits. There were 20 germplasm resources in group I, characterized by large fruit and thick peel. There were 20 germplasm resources in group II, characterized by a small fruit and thin pericarp and high FSR, DSR, and DSK. There were 127 germplasm resources in group III, which had medium fruit size, pericarp thickness, and high DSK and DKOC.

### 3.3. Genetic Diversity and Cluster Analysis

Genetic parameters based on 12 pairs of SSR primers are shown in Table 5. A total of 214 alleles were detected by 12 SSR primers, with an average of 17.833 alleles and 8.999 effective alleles. Shannon’s index ranged from 1.257 to 2.864, with an average of 2.302. The observed heterozygosity ranged from 0.867 to 1, with an average of 0.965. Expected heterozygosity ranged from 0.628 to 0.93, with an average of 0.850. The results for I, Ho, and He showed that the genetic diversity of 167 *C. oleifera* germplasm resources was high. The polymorphism information content (PIC) ranged from 0.592 to 0.927, with an average value of 0.836. The MAF ranged from 0.123 to 0.548, with an average of 0.238. Primers with a PIC higher than 0.5 indicated that that 12 pairs of primers had high polymorphism and could be used for diversity analysis and variety identification.

Based on phylogenetic tree analysis of Euclidean distance, the 167 *C. oleifera* germplasm resources were divided into three groups (Figure 3). Group I had four germplasm resources (GY86, GY137, GY88, and GY94); group II had four germplasm resources (GY124, GY117, GY110, and GY85); group III had 159 germplasm resources; and group III could be divided into four small groups. Group IIIa contained five germplasm resources (GY136, GY75, GY131, GY15, and GY9); Group IIIb only contained GY27; and Group IIIc contained 20 germplasm resources. Group IIId included the remaining 135 germplasm resources. The population structure of 167 germplasm resources was analyzed. The best classification population was two, and the Q values were higher than 0.6. Furthermore, the population results were relatively simple, but there were different classification combinations with the phylogenetic tree.

### 3.4. Construction of Core Collection

The core collections were constructed using PowerCore software and QGA software, respectively, according to different sampling proportions. The mean difference percentages (MDs) of the core collections were less than 20%, and the range coincidence rates (CRs) were greater than 80%, indicating that all core collections meet the conditions (Table 6). For core collections constructed by QGA software, MDs were in the range of 0–8.33%, with an average of 3.33%. CRs were in the range of 83.64–100%, with an average of 90.68%. The percentage differences of variance (VDs) were in the range of 8.33–75%, with an average of 38.33%. The change rates of coefficients of variation (VRs) were in the range of 116.87–128.66%, with an average of 121.92%. The trait retention rates (TRs) were in the range of 82.38–99.16%. The phenotypic diversity of the QGA-15, QGA-20, and QGA-25 core collections was rich. For core collections constructed by PowerCore software, MDs were in the range of 1.68–5.7%, with an average value of 3.57%. CRs were in the range of 84.61–92.18%, with an average value of 89.72%. VDs were in the range of 18.59–31.77%, with an average value of 25.87%. VRs were in the range of 109.43–116.03%, with an average value of 112.90%. TRs were in the range of 87.27–99.09%, and with an average value of 94.91%. The PowerCore-15, PowerCore-20, PowerCore-25, and PowerCore-30 core collections showed good phenotypic diversity.

Table 7 shows that for core collections constructed using PowerCore software, the Ne, I, Ho, He, and unbiased expected heterozygosity (uHe) of the core collections were all higher than in the original collection, and only Na was lower than in the original collection. We also constructed core collections using QGA software, only the Ho and uHe of QGA-15, QGA-10, and QGA-25 were slightly higher than in the original collection. In comparison, the PowerCore-25 can retain higher phenotypic traits of the original population and improve the variation and genetic diversity of the core collection. The mean value comparison and *t*-test between the PowerCore-25 and the original collection show no significant differences in 12 fruit traits (Table 8). The principal coordinate analysis of 167 *C. oleifera* germplasm resources showed that the core collection was evenly distributed in the original collection (Figure 4), indicating that the constructed core collection has a certain representativeness.

## 4. Discussion

### 4.1. Fruit Phenotypic Variation

Genetic diversity is mainly studied at the morphological, cytological, biochemical, and molecular levels to understand the genetic diversity of species [28], thus providing a scientific reference for the germplasm, conservation, and utilization of species. Phenotypic variation is caused by the interaction between genes and the environment [29], resulting in relatively rich diversity within and between species and populations. Morphological markers are the simplest genetic markers that can, to a certain extent, reveal genetic variation within or between species [8,30]. In morphological study on pecan (*Carya dabieshanensis* M. C. Liu et Z. J. Li) [27], apricot (*Prunus armeniaca* L.) [31] and melon (*C. melo L.*) [32], substantial differences were found in phenotypic traits. In this study, seven phenotypic traits were measured in 167 germplasm resources from Guizhou Province. The coefficient of variation between and within cultivars ranged from 11.79% to 61.76% and from 8.15% to 42.31%, respectively, indicating that the phenotypic variation of 167 *C. oleifera* germplasm resources was also extremely abundant. He et al. [8] studied 150 oil tea germplasm resources, and found that the phenotypic variation range among four fruits was 13.24–39.60%. The results were consistent with this study, with relatively less variation in FHD and FVD and large variation in FFW, PT, and SGN. Gao et al. studied *C. oleifera* in 18 areas in China, and found that the coefficient of variation of FHD and FVD was less than 20% [33]. However, the variation in fruit traits was substantial in this study, and the average coefficient of variation of seven traits was 36.75%. The results were consistent with Xie et al.’s study on the phenotypic traits of four *C. meiocarpa* and one *C. oleifera* plants: the coefficients of variation of five phenotypic traits ranged from 23.86% to 56.94% in this study [34]. The germplasm resources of oil tea in this study may include part of *C. oleifera* and *C. meiocarpa*. *C. meiocarpa* has the characteristics of thin skin and small fruit, and its fruit phenotype was significantly different from that of *C. oleifera* [35]. In this study, FFW, PT, and SGN of oil tea had a large variation range, traits which could be used as reference for the selective breeding for oil tea.

### 4.2. Correlation Analysis and Cluster Analysis

Association analysis can reveal the relationship between different traits [36]. In this study, the 12 traits showed significant correlation, and the results were consistent with the results based on 150 oil tea plants provided by He et al. [8]. FSR and DSR were negatively correlated with six phenotypic traits. Liang et al. [20] studied 40 *C. oleifera* plants from Guizhou and found that only FSR and PT were negatively correlated, while DSR was negatively correlated with FHD. The result was probably due to the fact that the 40 *C. oleifera* plants came from a unique climate in low, hot valleys, exhibiting specific genotypic performance. Furthermore, Lu Yang studied 45 superior *C. weiningensis* YK Li., and found no significant correlation among PT and other fruit traits and economic traits. These results may be because the studied materials belong to the thin-skinned type, and the variation among the characteristics was small [37]. The results also indicated that *C. oleifera* has great potential for genetic improvement and is important for crossbreeding.

The 167 germplasm resources were divided into three groups by 12 phenotypic traits. Group I had 20 germplasm resources with large fruit and thick pericarp, and their FVD, FHD, FFW, and PT were 33.05%, 21.73%, 113.78%, and 41.49% higher than those in other groups. There were 20 germplasm resources in group II, with small and thick fruit pericarp and a high seed rate. The fresh and dry seed rates were 41.34% and 33.38% higher than those in other groups. There ware 127 germplasm resources in group III, and the oil content rate of dry kernels in this group was 27.22% higher than in other groups.

### 4.3. Genetic Diversity and Genetic Structure Analysis

In this study, 167 *C. oleifera* germplasm resources showed high genetic diversity. Jia et al. [25] studied 18 *C. oleifera* germplasm resources, and found the average values of Na, Ho, and He were 9.1, 0.741, and 0.746, respectively. The results were consistent with our results, indicating that *C. oleifera* has high genetic diversity due to geographical isolation and late self-incompatibility [38]. Zhao et al. [39] studied 50 genotypes of *C. japonica* and *C. oleifera* with 21 SSR pairs, and found that the H, I and PIC of *C. oleifera* were 0.2089, 0.3324, and 0.4014, respectively. The genetic parameters were smaller than those in this study, possibly because fewer research samples was used.

Based on SSR molecular marker information clustering, 167 *C. oleifera* germplasm resources were divided into three groups. The clustering of 12 fruit traits also demonstrated three groups, but the two methods differed greatly regarding the classification groups. Li et al. studied 89 *C. camellia* genotypes and found that the same species could cluster together well, and some accessions that were grouped in the same cluster or subcluster had similar flower colors, but these groups can also exist in different colors [40]. Jan et al. studied 105 barley (*Hordeum vulgare* L.) genotypes and found that the genotypes clustered in three sub-clusters did not show any trait-specific relationship with each other [41]. There were differences between the clustering based on molecular marker information and the clustering of fruit traits in this study. These differences may be because the samples used in this study were collected from different geographical origins, and the fruit traits were influenced by both genes and the environment, leading to natural variations in morphology, physiology, structure, and gene expression [42]. It may also be related to the number of SSR primers; Velázquez-Barrera et al. studied the population structure of 118 cultivated pear trees using 12 SSRs and 18 SSRs, they found different results of population structure when they eliminated the higher percentile of null alleles and linked loci [43]. The results showed that SSR primers have a substantial influence on classification.

### 4.4. Construction of Core Collections

Core collections can preserve the genetic diversity of the original collection to the maximum extent [44]. In this study, the effective value parameters of fruit traits and the genetic parameters of molecular marker information were comprehensively compared. The results suggest that core collections generated using PowerCore-25 are relatively better core collections which can retain the favorable fruit traits and high genetic diversity of the original collection. Combining phenotypic traits with molecular marker information, a core collection constructed using PowerCore software has been reported for walnuts (*Juglans regia* L.) [45], which could improve the effective value parameters and genetic parameters of the core collection. The results were consistent with this study. Kim et al. [46] used PowerCore software to build the core collection of Korean apple, and the parameters of Ne, I, and He of the core collection were greatly improved compared with the original collection. For olive (*Olea europaea* L.) [47] and *Perilla* [12], core collections constructed using PowerCore retained high allele loci and trait characteristics. In phenotypic and genetic parameters, PowerCore was used to establish a walnut core collection with the total coverage of traits in the entire collection [48]. PowerCore software is convenient and efficient for building core collections, and can be used to lay a foundation for the subsequent management of the germplasm bank.

## 5. Conclusions

In this study, 12 pairs of primers were found to show high polymorphism, which can provide a reference for subsequent studies on the genetic diversity of *C. oleifera*. Germplasm resources of *C. oleifera* in Guizhou were found to have a high genetic diversity and can be used to improve breeding. However, the cluster analysis showed that the environment had a substantial influence on the phenotypic and economic traits of *C. oleifera*. Therefore, the environment should be taken into consideration when introducing and breeding *C. oleifera*. In addition, the core collection of phenotype traits, economic traits, and molecular marker information was constructed in this study, and was found to be conducive to the preservation and management of *C. oleifera*.

## Figures and Tables

**Figure 1 genes-13-02351-f001:**
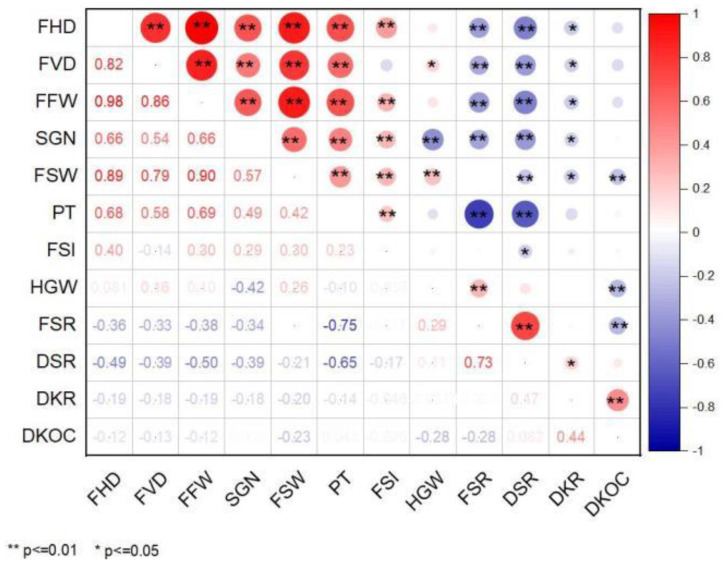
Correlation analysis of fruit traits. Note: In Figure 1, the lower left corner is the Pearson correlation value; in the lower right corner, **, represents a highly significant correlation, *p* ≤ 0.01; * represents a significant correlation, *p* ≤ 0.05.

**Figure 2 genes-13-02351-f002:**
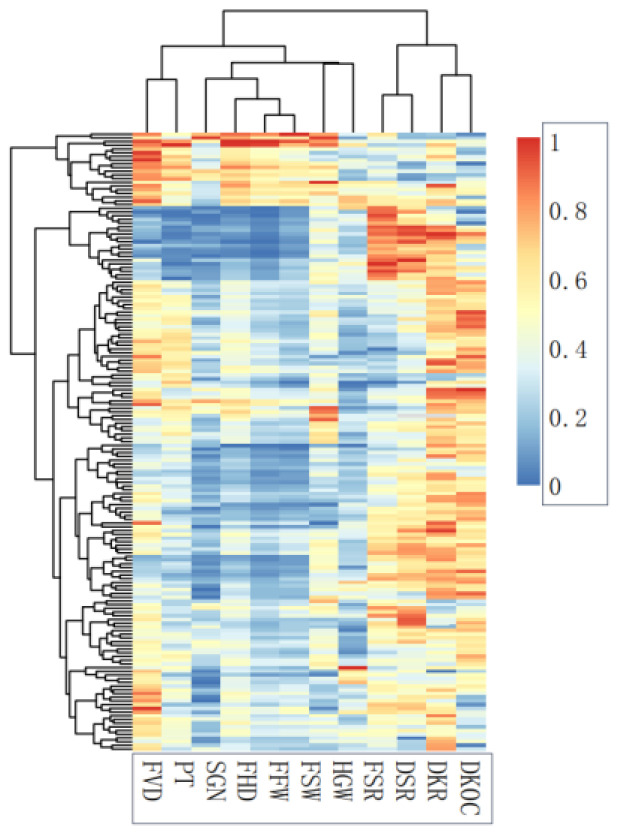
Clustering heat map of fruit traits.

**Figure 3 genes-13-02351-f003:**
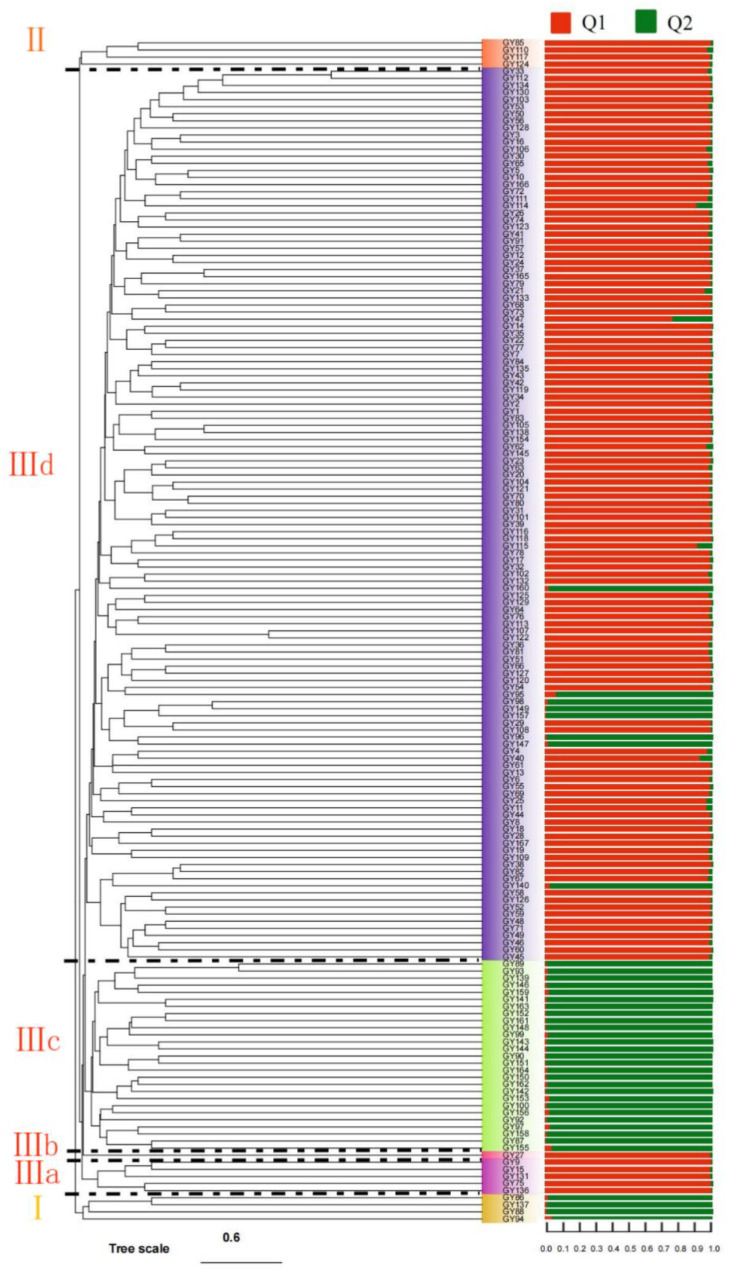
Phylogenetic tree and genetic structure of 167 *C. oleifera* germplasm resources based on SSR Markers. Note: The stacked bar charts represent the Q value of different assigned groups. Q1 and Q2 represent two groups when the genetic structure of 167 *C. oleifera* germplasm resources is divided into two groups.

**Figure 4 genes-13-02351-f004:**
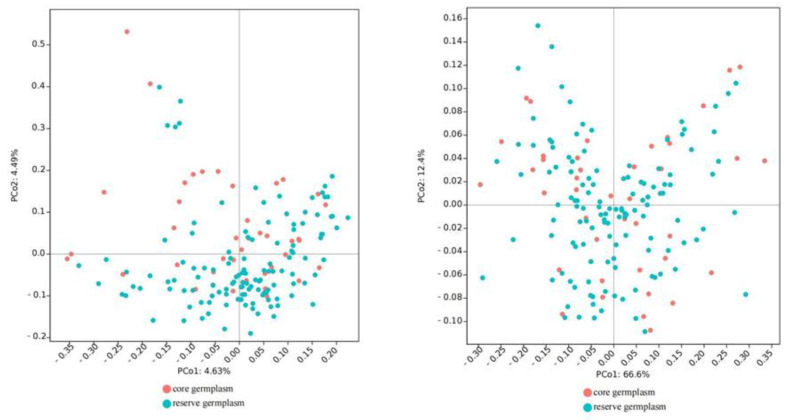
Principal coordinate analysis based on SSR 167 germplasm resources (**left**) and principal coordinate analysis based on fruit traits of 167 germplasm resources (**right**).

**Table 1 genes-13-02351-t001:** Information about the 167 *C. oleifera* germplasm samples collected.

Collection Place	North Latitude/ (°)	East Longitude / (°)	Altitude/m	Germplasm Code
Liping in Guizhou	26.3355	109.1819	438	GY85-164
Jinping in Guizhou	26.6096	109.3642	515	GY1-35
Yuping in Guizhou	27.3125	108.9194	536	GY165-167
Guizhou Academy of Forestry	26.5241	106.7426	1221	GY36-84

**Table 2 genes-13-02351-t002:** SSR primers and sequence information.

Marker Name	Forward (5′-3′)	Reverse (3′-5′)	Tm/ ℃	References
CoUg3417	CGGGAATCAAAAAGCTAGG	AGTGGGTGGTGAACACCAA	56	[24]
CoUg4931	CTTCATCTCCGTTTGCTCT	GACTTTGCCTCCTTTTGTG	55	[25]
CoUg7330	ACAACCATCTTCCTCTCCCC	CGTCGTCCCTGTTCACCTCT	60	[25]
CoUg11592	TACCGTGCTTGTGTATTCC	GTGTTTTGTGTTGCTGCCT	55	[24]
CoUg3402	ACTCTTGTGGGTGAATGTTG	GCTGGTAGGTTGGTTATGTT	56	[25]
CoUg5179	AATGGAGAATGAATGGACAG	GCAGAAAGTGATATTGGGTG	55	[25]
CoUg11169	GTCTGGTGGCGTTGCTTGCT	GTCTGCTGATCCGATGGCTG	62	[24]
CoUg17436	TTGAGGGTGAAGTCGATGA	AAGGAGTTGGTGAGTAGCA	55	[24]
CoUg8099	TGGGGATTGCTCAAAAGTGT	AGGGTGGCTGTGCTGGTATT	58	[24]
CoUg8134	CCAGAGCCAGGAGGAAGTA	GAGAGAGGGGGTAGAATGA	58	[24]
CoUg13753	CACATCATTAGGGTCGTTG	GGTTTTCACTCTTCAGCAG	55	[24]
CoUg4364	GTGGTCCTGGAGATCTGTCC	TTTCGCTCTATCCGTTGTTC	57	[25]

**Table 3 genes-13-02351-t003:** Analysis of phenotypic variance among different varieties and within varieties of germplasm resources of *C. oleifera* in Guizhou.

Phenotypic Traits	Mean ± SD	MS			F Value	
		Among Different Varieties	Within the Varieties	Random Errors	Among Different Varieties	Within the Varieties
FHD/mm	24.509 ± 4.646	442.992	26.111	6.901	64.081 **	2.435 **
FVD/mm	26.734 ± 4.544	398.788	24.009	7.517	52.933 **	2.357 **
FFW/g	9.388 ± 5.123	518.612	33.615	8.943	57.828 **	2.71 **
SGN	2.795 ± 1.726	45.910	4.002	1.446	31.735 **	2.184 **
FSW/g	4.195 ± 2.388	93.382	9.422	2.455	37.895 **	2.724 **
PT/mm	2.404 ± 0.873	17.038	0.483	0.226	75.322 **	2.071 **
FSI	0.92 ± 0.108	0.194	0.009	0.006	33.598 **	1.439 **

Note: ** extremely significant correlation, *p* < 0.01. FSI: fruit shape index, FSI = FHD/FVD.

**Table 4 genes-13-02351-t004:** Coefficient of variation, variance component, and phenotypic differentiation coefficient of *C. oleifera* in Guizhou.

Phenotypic Traits	Coefficient of Variation/ %		Variance Component			Percentage of Variance Component/ %		Phenotypic Differentiation Coefficient/ %
	among Different Varieties	within the Varieties	among Different Varieties	within the Varieties	Random Errors	among Different Varieties	within the Varieties	
FHD/mm	18.95	11.15	12.986	1.786	6.9013	59.92	8.24	87.91
FVD/mm	17.00	10.55	11.675	1.534	7.5173	56.33	7.4	88.39
FFW/g	54.56	30.14	15.108	2.294	8.9427	57.35	8.71	86.82
SGN	61.76	42.31	1.306	0.238	1.4455	43.68	7.95	84.60
FSW/g	56.92	36.42	2.616	0.648	2.4552	45.74	11.33	80.15
PT/mm	36.32	19.73	0.516	0.024	0.2261	67.35	3.13	95.56
FSI	11.79	8.15	0.006	0.000	0.0057	48.75	2.59	94.96

**Table 5 genes-13-02351-t005:** Genetic parameters based on 12 SSR in this study.

Marker Name	Na	Ne	I	Ho	He	PIC	MAF
CoUg11592	8	2.830	1.285	0.952	0.647	0.592	0.500
CoUg3402	6	2.689	1.257	0.867	0.628	0.590	0.548
CoUg3417	27	14.228	2.864	0.982	0.930	0.927	0.123
CoUg8134	19	10.662	2.573	0.994	0.906	0.903	0.186
CoUg5179	20	9.627	2.481	0.897	0.896	0.890	0.156
CoUg8099	14	4.085	1.751	0.915	0.755	0.736	0.398
CoUg17436	23	11.578	2.716	0.976	0.914	0.909	0.180
CoUg4364	19	10.267	2.541	1.000	0.903	0.895	0.153
CoUg4931	16	10.236	2.467	1.000	0.902	0.895	0.138
CoUg11169	18	10.043	2.508	1.000	0.900	0.892	0.174
CoUg13753	18	9.735	2.401	0.994	0.897	0.889	0.135
CoUg7330	26	12.008	2.767	1.000	0.917	0.911	0.171
Mean	17.833	8.999	2.301	0.965	0.850	0.836	0.238

Note: Na: number of different alleles; Ne: number of effective alleles; I: Shannon’s information index; Ho: observed heterozygosity; He: expected heterozygosity; PIC: polymorphism information content; MAF: major allele frequency.

**Table 6 genes-13-02351-t006:** Comparison of effective evaluation parameters of *C. oleifera* core collection based on traits.

Method	Sampling Rate (%)	Evaluation Parameters				
		Trait Retention Ratio (%)	Mean Difference (%)	Variance Difference (%)	Coincidence Rate of Range (%)	Rate of Variable Coefficient Change (%)
QGA	10	82.38	8.33	25	83.64	121.58
	15	93.77	0	41.67	87.03	123.96
	20	93.77	0	41.67	87.83	118.55
	25	99.16	8.33	75	100	128.66
	30	97.50	0	8.33	94.9	116.87
PowerCore	10	87.27	5.7	31.77	84.61	116.03
	15	94.55	3.11	30.21	88.92	115.99
	20	96.36	4.28	26.63	91.08	111.26
	25	97.27	3.08	22.14	91.82	111.77
	30	99.09	1.68	18.59	92.18	109.43

**Table 7 genes-13-02351-t007:** Comparison of genetic parameters of *C. oleifera* core collections based on SSR.

Method		QGA					PowerCore				
Sampling Rate	100%	10%	15%	20%	25%	30%	10%	15%	20%	25%	30%
Na	17.833	11.333	12.250	13.750	13.833	14.500	12.667	14.417	15.250	15.833	16.750
Ne	8.999	7.811	7.945	8.442	8.364	8.451	9.206	9.686	9.825	9.857	9.592
I	2.301	2.122	2.161	2.209	2.215	2.228	2.277	2.326	2.336	2.350	2.352
Ho	0.965	0.979	0.979	0.962	0.973	0.962	0.978	0.973	0.974	0.973	0.969
He	0.850	0.835	0.841	0.842	0.846	0.839	0.864	0.862	0.859	0.861	0.857
uHe	0.852	0.862	0.859	0.855	0.857	0.848	0.893	0.880	0.872	0.872	0.866

Note: Na: number of different alleles; Ne: number of effective alleles; I: Shannon’s information Index; Ho: observed heterozygosity; He: expected heterozygosity; uHe: unbiased expected heterozygosity.

**Table 8 genes-13-02351-t008:** Comparison of character parameters between the original, reserved, and core collections.

Traits	Original Collection		Core Collection		Reserve Collection		t1	t2
	Mean ± SD	Range	Mean ± SD	Range	Mean ± SD	Range		
FHD	24.22 ± 3.75	17.04–36.41	24.61 ± 4.45	17.5–36.22	24.09 ± 3.51	17.04–36.41	0.581	−0.298
FVD	26.57 ± 3.57	18.34–34.75	26.64 ± 4.02	18.65–34.31	26.55 ± 3.43	18.34–34.75	0.110	−0.055
FFW	9.07 ± 4.06	3.20–25.64	9.69 ± 5.02	3.36–24.89	8.87 ± 3.69	3.20–25.64	0.826	−0.433
SGN	2.72 ± 1.20	1.10–8.33	2.8 ± 1.23	1.10–6.4	2.69 ± 1.19	1.10–8.33	0.408	−0.198
FSW	4.10 ± 1.74	1.34–11.62	4.39 ± 2.21	1.51–11.62	4.01 ± 1.55	1.34–10.86	0.900	−0.48
PT	2.35 ± 0.72	0.89–4.67	2.44 ± 0.84	0.93–4.67	2.32 ± 0.68	0.89–3.99	0.730	−0.372
FSI	0.92 ± 0.08	0.74–1.16	0.93 ± 0.08	0.76–1.13	0.91 ± 0.08	0.74–1.16	0.868	−0.426
HGW	163.28 ± 63.54	48.69–444.58	171.4 ± 80.61	48.69–444.58	160.63 ± 57.03	59.02–337.88	0.693	−0.368
FSR	0.47 ± 0.09	0.26–0.69	0.47 ± 0.10	0.26–0.69	0.46 ± 0.08	0.27–0.67	0.188	−0.097
DSR	0.25 ± 0.05	0.12–0.38	0.25 ± 0.05	0.14–0.36	0.25 ± 0.05	0.12–0.38	−0.494	0.240
DKR	0.64 ± 0.06	0.46–0.76	0.63 ± 0.06	0.51–0.74	0.64 ± 0.06	0.46–0.76	−0.901	0.431
DKOC	0.42 ± 0.09	0.20–0.60	0.40 ± 0.10	0.21–0.60	0.42 ± 0.08	0.20–0.58	−1.306	0.651

Note: t1 represents the *t*-test value of each trait in the original germplasm collection and the core germplasm collection, and t2 represents the *t*-test value of each trait of core germplasm collection and reserved germplasm collection. HGW: weight of hundred grains; FSR: fresh seed rate; DSR: dry seed rate; DKR: dry kernel rate; DKOC: oil content rate of the dry kernel.

## Data Availability

Not applicable.

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
