# Peer review of "Genetic Diversity Analysis and Core Germplasm Collection Construction of Camellia oleifera Based on Fruit Phenotype and SSR Data"

_genes, 2022, doi:10.3390/genes13122351_

Round 1
Reviewer 1 Report
First and foremost, this manuscript needs an extensive editing of English language. It is very difficult for reviewers to provide feedback if the study is not presented clearly. For example, many sentences need to be revised and divided to a few short ones. Line 410-416 is an obvious problematic sentence in that group.
The plant material of this study is a collection of 167 Camellia oleifera germplasms. There is no other species involved. It is very confusing to read those parts talking about intraspecific and interspecific variation. I guess the authors were trying to present the fruit trait variation within a variety or among different varieties. This is just an example. Many language and word errors in the context must be corrected prior to an effective review.
Title is “…SSR data”, not “…SSR date”. It is also very hard to find how the SSR markers were developed in this study.
Author Response
Manuscript ID: genes-1953383
Title: Genetic diversity analysis and core germplasm collection construction of Camellia oleifera based on fruit phenotype and SSR data
Dear reviewers:
Thank you for your useful comments on our manuscript. We wish to give a sincere gratitude to referees for reviewing our paper carefully. We apologize for any inconveniences caused by these errors. We have modified the manuscript accordingly, and the response to the referees’ comments are listed point by point below:
Comment 1: First and foremost, this manuscript needs an extensive editing of English language. It is very difficult for reviewers to provide feedback if the study is not presented clearly. For example, many sentences need to be revised and divided to a few short ones. Line 410-416 is an obvious problematic sentence in that group.
Reply: Thanks for your comment and suggestion. we are very sorry for many language and word errors. We have edited the full paper for English. And we have supplemented the references for 12 SSR markers.
Comment 2: The plant material of this study is a collection of 167 Camellia oleifera germplasms. There is no other species involved. It is very confusing to read those parts talking about intraspecific and interspecific variation. I guess the authors were trying to present the fruit trait variation within a variety or among different varieties. This is just an example. Many language and word errors in the context must be corrected prior to an effective review.
Reply: Thanks for your comment and suggestion, we are very sorry for many language and word errors. We studied the fruit trait variation within a variety or among different varieties, we have checked and revised the language and word errors of the paper.
Comment 3: Title is “…SSR data”, not “…SSR date”. It is also very hard to find how the SSR markers were developed in this study.
Reply: Thanks for your comment and suggestion, we have revised it in the paper.

Reviewer 2 Report
The results of the work are interesting but the formatting of the text needs to be reviewed. Please change the caption of table 3 from "SSR primer information table" to "Genetic parameters based on 12 SSR used in this study". Otherwise, you can make two different tables, one for the information of the primers (name, sequences, references) and another one with the genetic parameters (Na, Ne, He, Ho, I, PIC, MAF, F). Anyway and for completeness of the information you should insert the effective number of alleles (Ne), the observed heterozygosity (Ho) and the Inbreeding coefficient (F). Please write in italics Camellia oleifera where not present.
Author Response
Thank you for your useful comments on our manuscript. We wish to give a sincere gratitude to referees for reviewing our paper carefully. Please see the attachment.

Reviewer 3 Report
The aim of this study is to study genetic diversity in Camellia oleifera germplasm, and to establish core collections.
I have some suggestions to authors:
Clear aim of study must be formulated at the end of introduction.
Germplasm studied must be described in materials and methods, and list of cultivars, hybrids, wild type plants etc. should be given as supplement material. Without this information, whole manuscript may be rejected.
Methods must be described, in particular molecular analysis, whole 2.3 section.
Camellia oleifera should be C. oleifera in the text after first mentioning.
Authors talk about differences between species, but they studied only one species - C. oleifera.
Discussion is very sparce, conclusions are too general.
Whole manuscript must be rewritten, English checked and carefully edited.
In general, I think manuscript should be rejected.
Author Response
Manuscript ID: genes-1953383
Title: Genetic diversity analysis and core germplasm collection construction of Camellia oleifera based on fruit phenotype and SSR data
Authors:
Dear reviewers:
Thank you for your useful comments on our manuscript. We wish to give a sincere gratitude to referees for reviewing our paper carefully. We apologize for any inconveniences caused by these errors. We have modified the manuscript accordingly,
and the response to the referees’ comments are listed point by point below:
Comment 1: Clear aim of study must be formulated at the end of introduction.
Reply: Thanks for your suggestion, we are very sorry for unclear aim. The aim of this study was to grasp the variation of fruit traits of C. oleifolia in Guizhou, and evaluate the genetic diversity and genetic relationship of C. oleifera germplasm resources in Guizhou province. In addition, constructing core collections of fruit traits and molecular marker information constructed, they laid the foundation for the breeding of C.oleifolia and the protection of germplasm resources
Comment 2: Germplasm studied must be described in materials and methods, and list of cultivars, hybrids, wild type plants etc. should be given as supplement material. Without this information, whole manuscript may be rejected.
Reply: Thanks for your suggestion. We have added the information of the research materials in Attached Table 1.
Comment 3: Methods must be described, in particular molecular analysis, whole 2.3 section.
Reply: Thanks for your suggestion. We have added DNA amplification (PCR reaction) system.
The fluorescence quantitative PCR reaction consists of a 20 µL system, including 17 µL gold Mix (green), 1 µL front primers, 1 µL back primers, and 1µL DNA template. The amplification procedures are listed as follows. First, pre-denaturation was performed at 98°C for 2 min. The second is the cycle stage. Samples were denatured at 98°C for 10s, deheated at ™10 s, extended at 72°C for 10 s, and cycled 35 times. The third stage is the extension stage. Samples were extended at 72°C for 5 min. For the capillary test, the mixing plate was heated at 95°C for 5 min with a metal bath heater and immediately put into an ice box at -20°C. The mixing plate was removed after cooling, centrifuged at 4000 rpm, thawed and mixed well. Then, Amplified SSR fragments were analyzed using a capillary electrophoresis sequencer ABI 3730 XL DNA analyzer.
Comment 4: Camellia oleifera should be C. oleifera in the text after first mentioning.
Reply: Thanks for your suggestion. We have revised it in the paper.
Comment 5: Authors talk about differences between species, but they studied only one species - C. oleifera.
Reply: Thanks for your comment, we are very sorry for word errors. We studied the fruit trait variation within a variety and among different varieties, we have checked and revised the word errors of full paper.
Comment 6: Discussion is very sparce, conclusions are too general. Whole manuscript must be rewritten, English checked and carefully edited.
Reply: Thanks for your comment. We have rewritten the Discussion and the full text, and we have edited the full text for English.

Reviewer 4 Report
The authors have investigated the genetic relationship of the germplasm of Camellia oleifera in Guizhou province, China, based on the phenotypic and genetic diversity, to construct the core germplasm collection resources. The authors identified high high genetic variation in Camellia oleifera based on 12 pairs of highly polymorphic SSR markers. The authors further constructed the core germplasm of Camellia oleifera based on both fruit traits and molecular markers. I believe this manuscript is presenting important data of the establishment of such a collection of germplasm in Camellia oleifera. However, I believe that the overall presentation of this manuscript, in particular, its English, needs a substantial improvement, if a revision is requested by the editor.
I have here provided an incomplete list of issues that should be addressed carefully by the authors in order to improve the manuscript. The authors need to realize that there are many more grammatical errors throughout the entire manuscript, making the reading and understanding of what exactly the authors were trying to say extremely challenging.
Title: change “date” to “data”
Abstract: I believe that the English of the Abstract needs major improvement.
The abstract needs to be significantly shortened by presenting the important results.
lines 13-15: grammatical errors
The genus name Camellia should be abbreviated as “C.” after its first use.
Line 34: What is QGA?
Introduction:
Again, the English of this section needs major improvement. The authors need to cite references for many statements in this section.
Line 43, a class of plants?
Line 44, grammatical error
Line 49-51, what is the logic of this statement? It is confusing and hard to know what the authors are trying to say here.
Lines 66-68, revise this sentence.
Lines 78-81, revise this sentence.
Lines 81 and 82, and…and…
Lines 83 and 84, the first letter of the specific epithet should not be capitalized..
Lines 84 and 85, revise this sentence.
Lines 89 and 90, Dai Yanhua?
Line 92, He Zhilong? Please make sure to use the correct format to cite an author…
Materials and Methods: again, the English of this section needs major improvement, I have seen a lot of broken sentences, making it very hard to understand what the authors are trying to say.
Line 112, the authors should provide the sources of the samples, to be presented in a table..
Lines 134-138, these four indices need an explanation first, something like “Four phenotypic characters were measures as follows: ”
Line 146, the authors should provide the primers and their sequences used in the qPCR.
Results:
Line 200, where are the 3 groups shown? Which figure or table?
Line 205: both captions of Figures 1 and 2 need more explicit explanations of what these figures are.
Line 235, the phylogenetic tree presented in Figure 3 needs to be enlarged to show the details of the branches.
Discussion:
As the important part of the manuscript, the Discussion needs to be significantly improved by citing and discussing relevant studies in the field, in addition to the English problems as shown throughout the entire manuscript.
Author Response
Thank you for your useful comments on our manuscript. We wish to give a sincere gratitude to referees for reviewing our paper carefully.Please see the attachment.

Round 2
Reviewer 1 Report
The manuscript has been substantially improved in writing, even though some sentences and paragraphs need further editing and polishing.
My major comment is that the results of this study can be better presented to provide more informative knowledge for C.oleifera. breeding and agricultural research. For example, how is the altitude of collection sites related to the fruit traits? Are there any climate or other environmental differences among those collection places in Guizhou? If so, are any of those factors potentially associated with fruit quality or a specific trait? What are the factors (genetic and environmental) that contribute to the production of a “superior” oil-tea variety? Which ones can be considered as the “elites” with great potentials among all your 167 varieties? In short, when presenting your results, please keep in mind how this study has adanced our knowledge to develope oil-tea as an economically-important oil plant in Guizhou.
Other comments and questions:
Table 1 change “Northern latitude” to “North latitude”. Alternatively, just present with “N” and “E” before those latitude and longitude numbers.
Table 3 change “Mead” to “Mean”. Please add units for each phenotypic trait if applicable.
Line 155 or Figure 1 What are phenotypic traits? What are economic traits? How do you define them?
Line 297-302 The discussion on classification discrepancy between SSR-based and fruit trait-based methods can be improved. Besides the G X E effects, the limited number of SSR markers, sample size and other factors should be considered. Please refer to more related plant quantitative genetics publications to interpret and discuss your results.
Some sentences obviously lack citations and the citation format in this text should be consistent.
Author Response
Thank you for your helpful comments on our manuscript. We sincerely thank the reviewers for carefully reviewing our paper.
We have revised the manuscript accordingly and have included a response to the reviewers' comments in the annex. Please refer to the attachment.

Reviewer 3 Report
Dear Authors,
Thank you for taking into account my remarks, however, I still miss some information on plant genetic material studied. You should clarify if studied plants are cultivars, hybrids, wild plants, etc. Also, English language must be improved, and there are still editing corrections required.
Author Response
Thank you for your useful comments on our manuscript. We wish to give a sincere gratitude to referees for reviewing our paper carefully. We apologize for any inconveniences caused by these errors. We have modified the manuscript accordingly, and the response to the referees’ comments are listed point by point below:
Comment 1: Thank you for taking into account my remarks, however, I still miss some information on plant genetic material studied. You should clarify if studied plants are cultivars, hybrids, wild plants, etc.
Reply: Thanks for your comment and suggestion. All of our materials are cultivars, include clone, family, etc. The clone varieties are grafted from buds of elite trees, the family varieties are reproduced from the seeds of elite trees.
Comment 2: Also, English language must be improved, and there are still editing corrections required.
Reply: Thanks for your comment and suggestion, we are very sorry for many language errors. We have checked and revised the language and word errors of the paper.

Reviewer 4 Report
I appreciate very much the efforts that the authors have devoted to improving their manuscript. I have no major problems but again, I hope that the authors should sharpen the English of this manuscript.
Author Response
Thank you for your useful comments on our manuscript. We wish to give a sincere gratitude to referees for reviewing our paper carefully. We apologize for any inconveniences caused by these errors.
Comment 1: I appreciate very much the efforts that the authors have devoted to improving their manuscript. I have no major problems but again, I hope that the authors should sharpen the English of this manuscript.
Reply: Thanks for your comment and suggestion, we are very sorry for many language errors. We have checked and revised the language errors of the paper.
